# Effects of Thinning on the Growth and Relative Change in the Diameter of a Mahogany Plantation

**Ping-Hsun Peng** [1,2], **Hsiao-Lung Pan** [2], **Sheng-Lin Tang** [2], **Chih-Ming Chiu** [3], **Hua-Le Chiang** [2], **Yu-Xuan Zhan** [4], **Yi-Ta Hsieh** [5] and **Jan-Chang Chen** [4,*]

[1] Graduate Institute of Bioresources, National Pingtung University of Science and Technology, Pingtung 91201, Taiwan; bspeng@tfri.gov.tw

[2] Taiwan Forestry Research Institute, Council of Agriculture (COA), Taipei City 10066, Taiwan; hlpan@tfri.gov.tw (H.-L.P.); daisy@tfri.gov.tw (S.-L.T.); clareman@tfri.gov.tw (H.-L.C.)

[3] Taipei Forestry Specialist Association, Taipei City 10053, Taiwan; chihming1314@gmail.com

[4] Department of Forestry, National Pingtung University of Science and Technology, Pingtung 91201, Taiwan; b10312050@g4e.npust.edu.tw

[5] General Research Service Center, National Pingtung University of Science and Technology, Pingtung 91201, Taiwan; blue90234570@mail.npust.edu.tw

* Correspondence: zzzjohn@g4e.npust.edu.tw

**Abstract:** Honduras mahogany (*Swietenia macrophylla* King) is an important forestry tree species in low altitude areas in central and southern Taiwan and has good potential for sustainable forestry in tropical regions. The aim of this study was to understand changes in the diameter at breast height (DBH) and stand structure of large-leafed mahogany. A lower layer thinning experiment was conducted in 2011 in a 14-year-old mahogany plantation in Guanmiao, Tainan City, Taiwan. Four zones of heavy, moderate, and light thinning, as well as a control were established and DBH surveys were conducted in 2011, 2012, 2013, 2015, and 2017 at tree ages of 14, 15, 16, 18, and 20 years, respectively. The DBH trend was observed using simple linear regression with continuous slope values and the Weibull density function was used to match the distribution of diameter classes and compare the average DBH growth under different thinning treatments. The results showed that the growth of small diameter trees remained slow after felling, whereas medium-intensity thinning could result in a similar increase in DBH for larger diameter trees within a certain period. The stand structure remained skewed ($c < 3.6$) six years after harvesting and spatial allocation needed to be re-adjusted to alleviate competition pressure. The mean periodic growth of a single tree DBH after thinning was significantly different from that of the un-thinned trees at tree ages of 16, 18, and 20 years ($p$ value $< 0.05$). However, the difference between thinning treatments was not significant and the effect of moderate thinning and intensive thinning was a similar in terms of promoting the mean periodic growth of single wood.

**Keywords:** thinning intensities; DBH; Weibull distribution

## 1. Introduction

Honduras mahogany (*Swietenia macrophylla*) is native to Central America. However, this species has thrived in Taiwan since it was introduced and has become an important forestry species in low altitude areas in central and southern Taiwan. Statistical data of general forestry area released by the Forestry Bureau from 1995 to 2019 shows the cumulative Honduras mahogany plantation area in Taiwan to be 1322.64 ha. This species is also a good shade tree due to its dense foliage, is popular within residential gardens and streets, and the extracts of the branches and leaves have anti-inflammatory properties [1] and can be used to treat hepatitis C [2]. The timber of this tree is also valued for the production of furniture, high-end handicrafts, yachts, and musical instruments [3], and therefore this species plays

an important role in the tropical forest timber market. Due to its biological and commercial properties, Honduras mahogany forestry has great potential for the development of large-scale sustainable forestry in tropical regions under the application of appropriate silviculture practices [4]. In response to initiatives for a green environment and a reduction in carbon emissions by afforestation, the Taiwan Kagome company planted trees in the Guanmiao area of Taiwan from 1996 to 1999. The resulting total afforestation area was 100 ha and large-leaved mahogany dominated the afforested area (60 ha), *Araucaria cunninghamii*, *Pterocarpus indicus*, and *Machilus zuihoensis* (40 ha). The afforested area, in combination with the surrounding forest environment in Taiwan, has become an important recreational place for residents. Thinning is considered to be an effective method of managing forest growth and productivity [5]. Thinning has comprehensive impacts on stand structure, reducing forest fire risk, and promoting forest growth. Therefore, thinning has benefits for vegetation, the landscape, recreational values, wildlife, and the economy [6,7]. Since forest growth is an ongoing dynamic process, it is necessary to identify the tree species-specific optimum age for thinning and nurturing, particularly in young stands in which an increase in DBH correlates with thinning intensity [8]. The management of stand density in a timely manner allows optimal growth benefits to be attained [9–14]. Reducing the horizontal spacing between adjacent trees allows the remaining trees to benefit from increased growing space and lowered competition, thereby contributing to increased DBH [15]. Therefore, the removal of slow-growing, poorly-shaped trees can increase stand productivity and growth, resulting in larger and better quality stands at harvest [16,17].

The present study aimed to understand the changes in DBH and stand structure of large-leaved mahogany. A low thinning trial was conducted in 2011 in a 14-year-old large-leaved mahogany plantation in Guanmiao, Tainan City, Taiwan over six years (2011, 2012, 2013, 2015, and 2017 representing tree ages of 14, 15, 16, 18, and 20, respectively). The objectives of the study were to: (1) observe changes in DBH of retained trees under different thinning intensities; (2) simulate the diameter class distribution using the Weibull probability density function, and; (3) investigate the changes and differences in the periodic average growth of stand DBH according to the thinning intensity. It is hoped that the results of the present study can assist in understanding how DBH growth is affected under different thinning rates before reaching the end of the afforestation period, in accordance with the 20-years base period of the "Plain Afforestation" policy in Taiwan.

## 2. Study Area, Materials, and Methods

### 2.1. Study Area

The study area of the current study was the Guanmiao Farm in Guanmiao, Taiwan, operated by the Kagome Corporation. The area has an altitude of 30–70 m and a slope of 12°–45° and the soil texture of the region is loamy. Gridded observations of temperature and rainfall in the study area from 1996 to 2015 were provided by the Taiwan Climate Change Information and Adaptation Knowledge Platform (TCCIP). The average annual temperature of the study area was 24.3 °C whereas the average annual total rainfall was ~2067 mm, with the rainy season occurring between May–September. Figure 1 shows the location and climate of the study area.

### 2.2. Data Collection

Survey data indicated that, before thinning, the average number of mahogany plants, DBH, height, and storage volume were $1718 \pm 242$ ha$^{-1}$, $14.9 \pm 1.7$ cm, $12.4 \pm 0.9$ m, and $210.7 \pm 50.7$ m$^3$, respectively. The forestation block was a monoculture, and part of the area contained a mix of vegetation habitat consisting of bamboo forest, miscellaneous trees, erosion ditches, lowlands, or adjacent buildings and roads.

The optimal thinning rate should aim to improve the health of the forest, facilitate the multiple benefits of afforestation, including carbon reduction, recreation, and economic benefits, and should minimize the aesthetic impacts of thinning. Therefore, depending on the forest and the adjacent environment, the planned thinning rate was 9–30% according

to the number of trees for lower layer thinning. The thinning method implemented was lower thinning, with a priority on the thinning of oblique trees and trees with broken or forked trunks, followed by the thinning of intermediate(or co-dominant) trees. This present study established four sample plots, each with an area of 0.05 ha, which were categorized according to the number of retained plants into: (1) heavily-thinned trees (~800 trees ha$^{-1}$); (2) moderately-thinned trees (1000 trees ha$^{-1}$); (3) lightly-thinned trees (1200 trees ha$^{-1}$); and (4) the un-thinned (1700 trees ha$^{-1}$). The sample plots were surveyed both before and after thinning. Table 1 shows the mean diameter at breast height (DBH) of the sample plots.

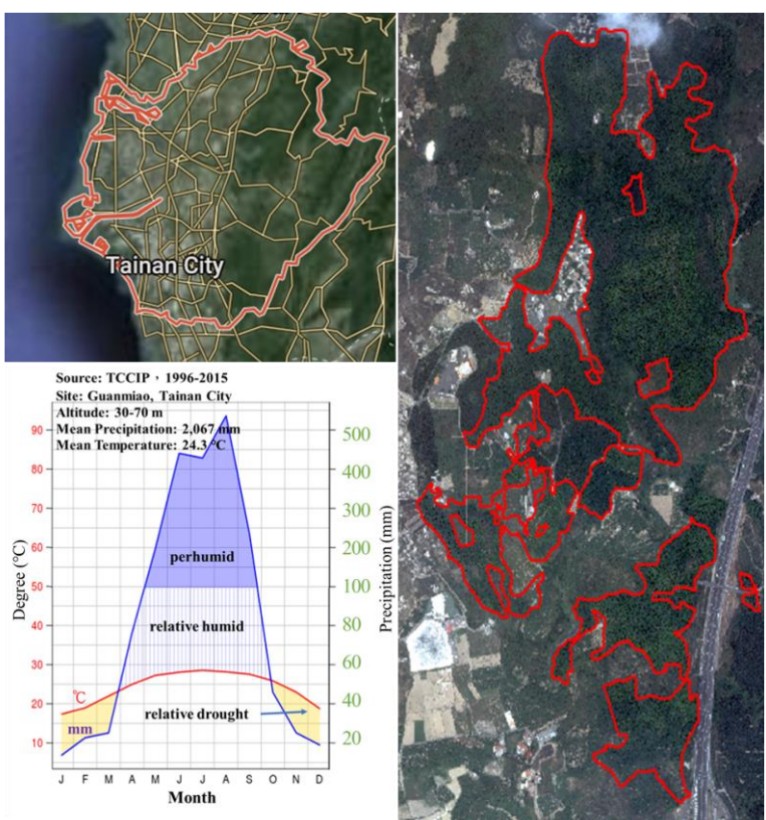

**Figure 1.** Map of the Guanmiao Farm, Guanmiao, Taiwan showing the afforested area of Honduras mahogany and the climograph.

**Table 1.** Average diameter at breast height (DBH; cm) of four plots of Honduras mahogany in the Guanmiao Farm, Guanmiao, Taiwan before and after thinning.

| Thinning Intensity | Tree Density (Trees ha$^{-1}$) | Pre-Thinning | Post-Thinning | Years after Thinning | | | |
|---|---|---|---|---|---|---|---|
| | | | | 1 | 2 | 4 | 6 |
| Heavy | 800 | 17.1 ± 6.8 | 19.6 ± 6.3 | 20.9 ± 6.5 | 21.5 ± 6.8 | 22.2 ± 7.2 | 23.4 ± 7.5 |
| Moderate | 1000 | 14.5 ± 4.6 | 16.4 ± 4.3 | 17.5 ± 4.3 | 18.1 ± 4.5 | 18.7 ± 4.8 | 19.7 ± 5.2 |
| Light | 1200 | 15.3 ± 4.9 | 16.7 ± 4.3 | 17.4 ± 4.8 | 17.9 ± 5.2 | 18.7 ± 5.8 | 19.6 ± 6.2 |
| Un-thinned | 1700 | 14.3 ± 5.2 | 14.3 ± 5.2 | 14.9 ± 5.4 | 15.2 ± 5.6 | 15.8 ± 5.7 | 16.9 ± 6.5 |

*2.3. Data Analysis*

In the present study, no survey data on the actual tree height and relative crown position were available. Therefore, it was assumed that the trees with DBH were relatively larger than the other trees within the stand and were dominant in the upper layer, whereas the opposite was true for growth-inhibited trees in the lower layer. After identifying the slope value of the linear regression equation, the continuous variation curves were plotted to identify the DBH growth rates under different diameters of retained wood.

The present study used a three-parameter Weibull probability density function to simulate the dynamic thinning process related to changes in stand structure. The minimum DBH at the time of inventory was applied as the value of the *a* parameter of the maximum likelihood estimator (MLE) and Newton's method was used to solve the values of parameters *b* and *c*. The goodness-of-fit was measured by the Kolmogorov-Smirnov Test (K-S test), with a critical value of $\alpha = 0.05$. Equations (1)–(3) mathematically define the above methods.

$$f(x) = \left(\frac{c}{b}\right) \times \left(\frac{x-a}{b}\right)^{c-1} \times exp\left[-\left(\frac{(x-a)}{b}\right)^{c}\right], \tag{1}$$

In Equation (1), *x* is the diameter at breast height (DBH), *a* is the location parameter, *b* is the scale parameter, and *c* is the shape parameter.

$$D_n = Max\left\{D_n^+, D_n^-\right\},$$

$$D_n^+ = Max_{1 \le i \le n}\left\{F_i - \hat{F}_i\right\} \tag{2}$$

$$D_n^- = Max_{1 \le i \le n}\left\{F_{i-1} - \hat{F}_i\right\}$$

$$D_a = \sqrt{\frac{-\ln\left(\frac{1}{2}a\right)}{2n}} \tag{3}$$

In Equations (2) and (3), $D_n$ is the critical value, $\alpha$ is the significance level, and *n* is the number of trees.

## 3. Results and Discussion

### 3.1. Effect of Thinning on the Growth and Relative Change of Single Wood DBH

As shown in Figure 2, the present study compiled the DBH values of surveyed standing timber in years one, two, four, and six after felling using simple linear regression analysis. As shown in Figure 3, the slopes of the linear regressions were plotted to compare the relative rate of change in single timber DBH growth under different thinning treatments. The slope will remain unchanged if trees of different diameters maintain the same DBH growth rate or if there is a complete stagnation of growth. In contrast, the growth of the lower layer is usually slow or stagnant in a stand under competitive pressure, whereas the upper layer shows better growth. The slopes of the un-thinned, light, moderate, and heavy thinned treatments were 1.08–1.15, 1.20–1.40, 1.12–1.27, and 1.11–1.21, respectively, with the rank of the treatments according to the regression slope being: light thinning > moderate thinning > heavy thinning > un-thinned. The treatments maintained a stable increasing trend in DBH until the sixth year after logging, with the highest and lowest DBH values for light thinning and the un-thinned area, respectively.

Ref [15] illustrated the growth of residual trees using the height-DBH relationship for the two post-felling periods (1997, 2009). Their results showed a higher increase in DBH after thinning for large trees than for small trees, which they attributed to the higher inhibition of small trees by shading [18,19]. Similarly, a study by [20] suggested that, in terms of the effect of the relative vertical position of the crown in the canopy on the volume growth, the canopy of a standing tree would experience optimal radial increase if located in the upper part of the canopy, and as such the optimal radial increase can be achieved by restricting thinned residual wood in the upper part of the canopy.

Although trees of the same size differ greatly in growth rate, larger trees have better access to light and nutrients. Therefore, there remains a significant positive correlation between tree growth and tree size [21]. Four observations of the present study showed that thinning intensity was inversely related to the change in the slope values of DBH-height regressions. In other words, increased thinning reduced the change in slope, whereas light thinning increased the change in slope, which can be attributed to the following factors: (1) Under light thinning, some small diameter trees remain inhibited due to insufficient

access to space and resources, whereas medium- and large-diameter trees grow faster. This results in an increased variation in DBH within four years of thinning. (2) The slope of the medium- and intensely-thinned areas decline after the second year, presumably because of the increasing closure of the canopy, thereby inhibiting the growth of remaining small-diameter trees. Therefore, it is believed that increasing the intensity of thinning promotes a similar DBH growth of remaining trees within a certain period.

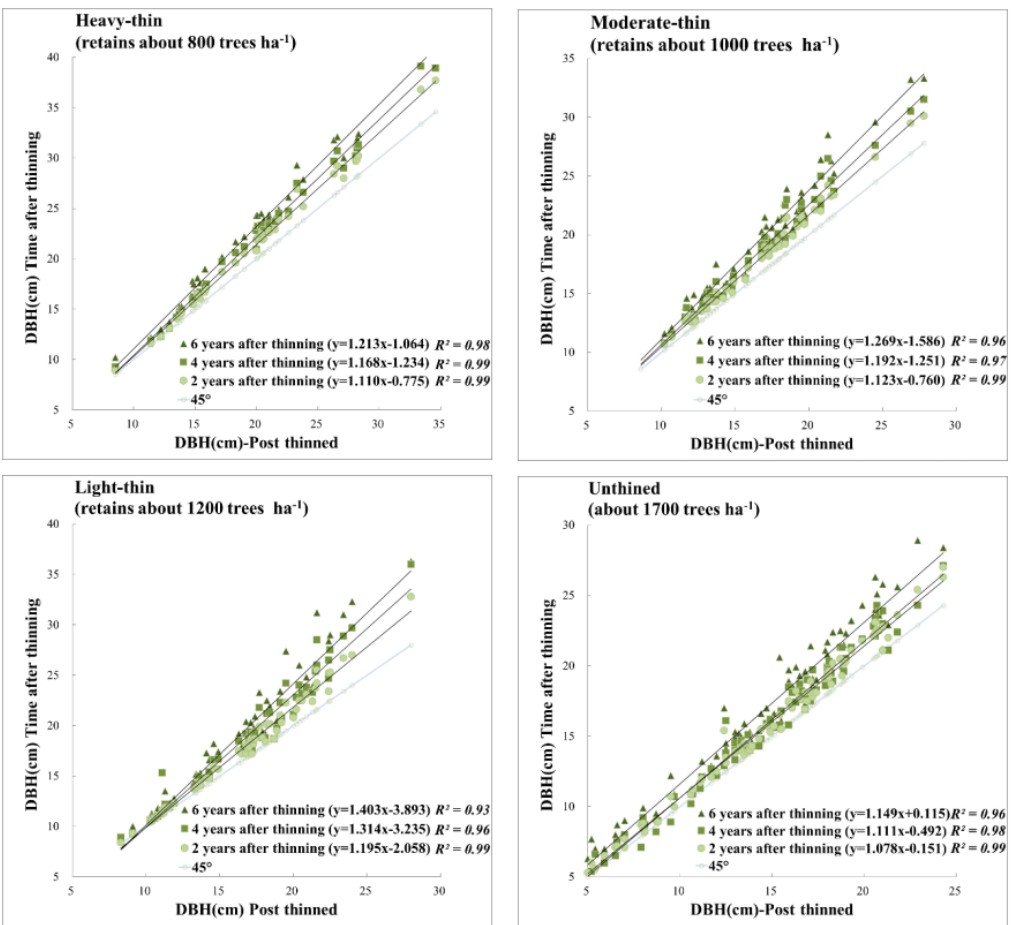

**Figure 2.** Scattering of DBH at 2–6 years under different thinning intensities of 14-year old mahogany Note: 45° line is the year after thinning (*X*-axis)—years after thinning (*Y*-axis).

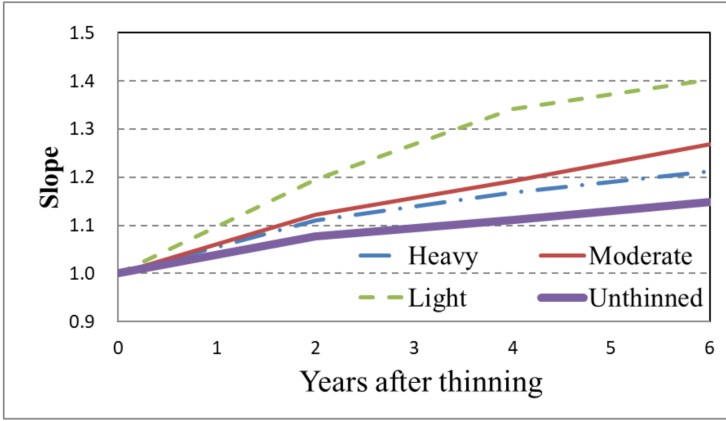

**Figure 3.** Variation of DBH growth slope during 6 years of 14-year old mahogany with four thinning treatments.

### 3.2. Effects of Thinning on Stand Structures

The Weibull probability density function was first applied as a diameter distribution model and is used in the analysis and testing of reliability and longevity. This function is characterized by a stretchy, closed curve shape, with associated extrapolated parameter values related to stand characteristics, such as stand age, stand mean height and density, and it can be used to predict the diameter distribution of trees under specific conditions [22,23]. The goodness-of-fit was measured by the Kolmogorov-Smirnov Test (K-S test) with a critical value of $\alpha = 0.05$. An absolute maximum difference $(D_n)$ in the cumulative frequency between observations and theoretical values of less than the critical value $(D_{0.05})$ indicated that the diameter distribution of the stand was consistent with the Weibull distribution and vice versa [22,24].

The present study applied the K-S test to the fitted estimates of parameters at the six time points to estimate the goodness-of-fit. As shown in Table 2, the results indicated that the $D_n$ values of all stands for the four treatments in six time points were all less than the critical value $(D_{0.05})$. Therefore, the Weibull probability density function was found to be suitable for estimating the diameter distribution of stands at different periods. The relative distribution of DBH and the predicted number of plants per hectare were obtained from the estimated parameters [25]. The definitions of the value *a*, *b*, and *c* can be found in related studies [22,24,26–32]. In this study, as shown in Table 2 and Figure 4, the values of *a* of the three thinned plots all increased after thinning, and the starting point of the diameter distribution shifted to the right to a larger value. Related studies have found that trees with a smaller DBH dominated stands in which less intense thinning was used to remove fallen trees. Therefore, the value of the *a* parameter is larger in stands subjected to intense thinning in which trees with a small DBH are removed, whereas the *a* parameter has a smaller value in stands which are lightly thinned [33]. In the present study, some trees with a small DBH and potential for growth were retained during thinning, resulting in an even distribution of trees. Therefore, although the value of *a* increased after felling, there was no increase in the degree of variation in *a* among treatments with increasing thinning intensity. The values of *a* maintained a small growth among treatments from one to six years after thinning, except for trees of the smallest DBH in the moderately-thinned treatment in which there was a large change in the values of *a* one year after thinning due to post-thinned lodging.

**Table 2.** Weibull parameter estimates and Kolmogorov-Smirnov (K-S) test results before and after thinning of Honduras mahogany in the Guanmiao Farm in Guanmiao, Taiwan (ns: non-significant in KS).

| Time | Thinning Intensity Class | Weibull | | | K-S Test | |
|---|---|---|---|---|---|---|
| | | *a* | *b* | *c* | $D_n$ | $D_{0.05}$ |
| Pre-thinned | Heavy | 4.9 | 13.55 | 1.9 | 0.095 [ns] | 0.220 |
| | Moderate | 6.0 | 9.68 | 2.4 | 0.089 [ns] | 0.175 |
| | Light | 4.0 | 12.89 | 2.32 | 0.090 [ns] | 0.185 |
| | Un-thinned | 2.7 | 13.54 | 2.51 | 0.071 [ns] | 0.178 |
| Post-thinned | Heavy | 8.5 | 12.30 | 1.72 | 0.106 [ns] | 0.264 |
| | Moderate | 7.1 | 10.66 | 2.35 | 0.082 [ns] | 0.226 |
| | Light | 8.3 | 9.92 | 1.96 | 0.102 [ns] | 0.214 |
| | Un-thinned | 2.7 | 13.54 | 2.51 | 0.071 [ns] | 0.178 |

**Table 2.** *Cont.*

| Time | Thinning Intensity Class | Weibull | | | K-S Test | |
| | | $a$ | $b$ | $c$ | $D_n$ | $D_{0.05}$ |
|---|---|---|---|---|---|---|
| One year after thinning | Heavy | 8.7 | 13.57 | 1.87 | 0.107 [ns] | 0.272 |
| | Moderate | 10.5 | 7.90 | 1.59 | 0.098 [ns] | 0.231 |
| | Light | 8.4 | 10.39 | 1.88 | 0.085 [ns] | 0.214 |
| | Un-thinned | 3.1 | 13.80 | 2.43 | 0.074 [ns] | 0.178 |
| Two years after thinning | Heavy | 8.9 | 13.99 | 1.83 | 0.108 [ns] | 0.272 |
| | Moderate | 10.7 | 8.34 | 1.55 | 0.104 [ns] | 0.231 |
| | Light | 8.4 | 10.87 | 1.83 | 0.078 [ns] | 0.214 |
| | Un-thinned | 3.2 | 14.05 | 2.36 | 0.073 [ns] | 0.178 |
| Four years after thinning | Heavy | 9.3 | 14.37 | 1.77 | 0.108 [ns] | 0.272 |
| | Moderate | 10.8 | 8.96 | 1.56 | 0.101 [ns] | 0.231 |
| | Light | 8.9 | 11.14 | 1.71 | 0.085 [ns] | 0.214 |
| | Un-thinned | 3.4 | 14.37 | 2.35 | 0.066 [ns] | 0.179 |
| Six years after thinning | Heavy | 10.2 | 14.73 | 1.76 | 0.133 [ns] | 0.276 |
| | Moderate | 11.6 | 9.05 | 1.49 | 0.110 [ns] | 0.231 |
| | Light | 8.9 | 12.16 | 1.76 | 0.074 [ns] | 0.216 |
| | Un-thinned | 4.3 | 14.57 | 2.29 | 0.062 [ns] | 0.182 |

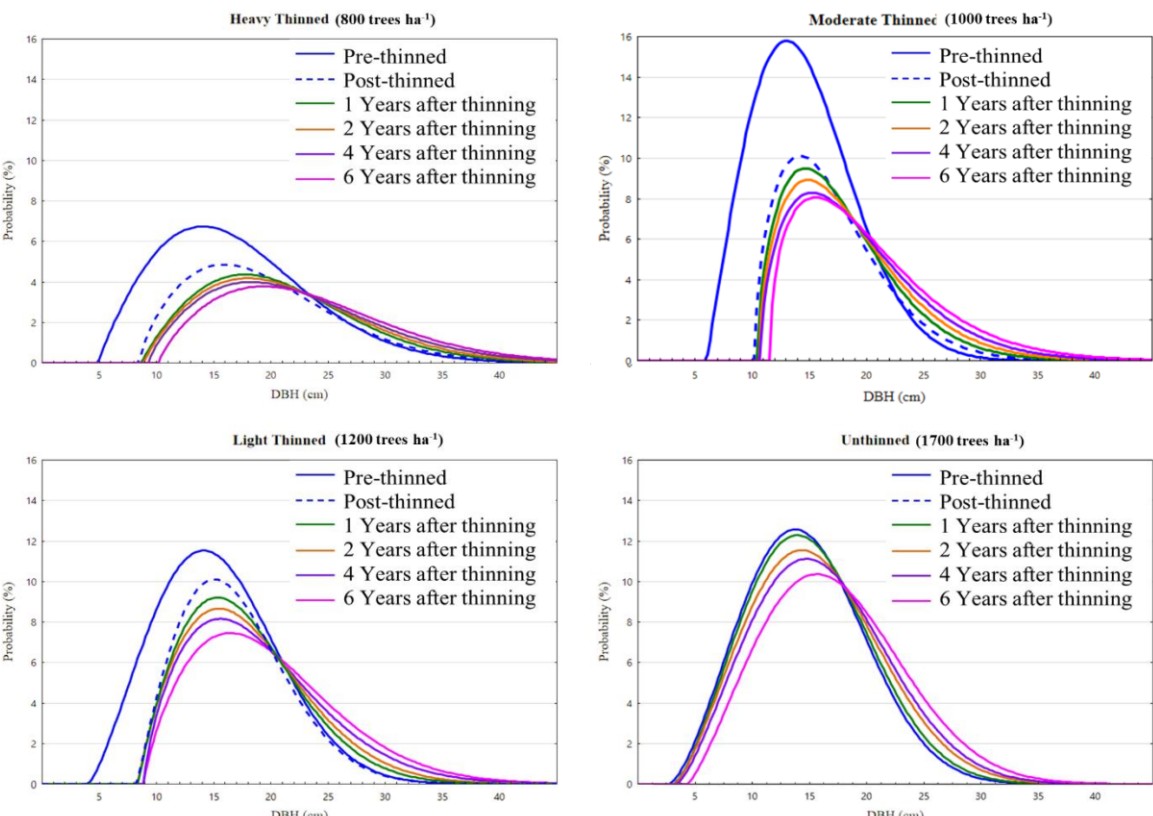

**Figure 4.** The distribution of diameter at breast height (DBH) among different thinning treatments of Honduras mahogany of an age of 14 in the Guanmiao Farm in Guanmiao, Taiwan before and after thinning.

The present study summed the values of the *a* and *b* parameters. In the sixth year after logging, the rank of the different thinning treatments in terms of the summed parameters was: intensely thinned (24.93) > moderately thinning (20.92) ≈ light thinning (21.06) > un-thinned (18.87). The rank of the thinning treatments in terms of *a* + *b* in the 6th year after logging was: intensely thinned (4.13) > moderately thinned (3.10) > light thinned (2.84) > un-thinned (2.63). Both the total and difference values increased with the increase in thinning intensity. The range of scale increased with the increase in thinning intensity, consistent with the results of previous studies [4,6,9,15]. The *c*-values in all six periods of the simulation maintained a positive skewed distribution of <3.6. The continuous decrease in *c*-values may indicate that some silviculture trees experienced stagnant or slow growth, resulting in an increase in the number of relatively small diameter silviculture trees.

### 3.3. Effects of Thinning on Average DBH among Individual Growth Periods

As shown in Table 3, the differences in average DBH among individual growth periods among the different thinning treatments was determined by one-way analysis of variance (ANOVA) and Fisher's least significant difference (LSD), showing a significant difference between thinning treatments and the un-thinned between two and six years after thinning ($p < 0.05$). However, there were no significant differences between the three thinning treatments, and the changes in the average DBH among individual growth periods were similar. Figure 5 compares the percentage change in DBH among individual growth periods for each treatment relative to the un-thinned. Two years after thinning, the DBH among moderately-thinned trees was ~10% higher than that of heavily-thinned trees. Four years after thinning, the rank of the different thinning treatments in terms of relative increases in DBH was: moderately-thinned trees = heavily-thinned trees > lightly-thinned trees > the un-thinned. The benefits of thinning for increasing DBH over individual DBH growth periods gradually decreased with increasing years after thinning, and the differences between the three treatments gradually decreased.

**Table 3.** Changes in diameter at breast height (DBH; cm) among different thinning treatments of Honduras mahogany of an age of 14 in the Guanmiao Farm in Guanmiao, Taiwan after thinning.

| Thinning Intensity Class | Years after Thinning | | | |
| --- | --- | --- | --- | --- |
| | 1 | 2 | 4 | 6 |
| Heavy | 0.85 ± 0.6 | 0.72 [a] ± 0.4 | 0.54 [a] ± 0.3 | 0.54 [a] ± 0.3 |
| Moderate | 0.92 ± 1.4 | 0.77 [a] ± 0.8 | 0.55 [a] ± 0.5 | 0.53 [a] ± 0.4 |
| Light | 0.73 ± 0.6 | 0.60 [ab] ± 0.5 | 0.51 [a] ± 0.5 | 0.48 [ab] ± 0.4 |
| Un-thinned | 0.61 ± 0.5 | 0.49 [b] ± 0.3 | 0.36 [b] ± 0.2 | 0.39 [b] ± 0.2 |
| *F value* | 1.82 | 3.47 * | 4.08 ** | 2.81 * |

Note: 1. Post-thinned DBH represents the mean annual increment = (DBH after thinning survey-DBH after thinning)/Survey interval years; 2. [a, b] represent significant different in Fisher's least significant difference (LSD) test ($\alpha \leq 0.05$); 3. * ($p < 0.05$), ** ($p < 0.01$).

It has been suggested a continual decline in the DBH growth rate to the same level as that of un-thinned sites can be observed when canopy closure is almost complete [8,15]. It remains necessary to assess the timing of the increase in growth according to the time of canopy depression. For example, a study by [8] of a stand containing six tree species of trees (ash, aspen, birch, oak, pine, and spruce) of 30–40 years old subjected to a 25–35% volume thinning rate observed a maximum growth rate among 2–3-year old trees after thinning, with European white oak (*Quercus robur*) and European poplar (*Populus tremula*) showing the best response (40–50%), whereas the remaining species showed a 30% rate of increase. A study of a Japanese larch (*Larix kaempferi*) plantation by [15] two to three years after logging observed a maximum growth rate in moderately-thinned and intensely-thinned trees of 26% and 24%, respectively, whereas the growth rates declined to the same rate as that of un-thinned land seven to eight years after logging. In an analysis of thinning data for 14-year old Taiwan fir trees, [30] showed that the relative growth rate of retained

trees with a diameter class of ≥20 cm reached a maximum in the fifth year after felling, following which the growth rate showed an obvious decline between five and seven years after felling, followed by a gradual decline.

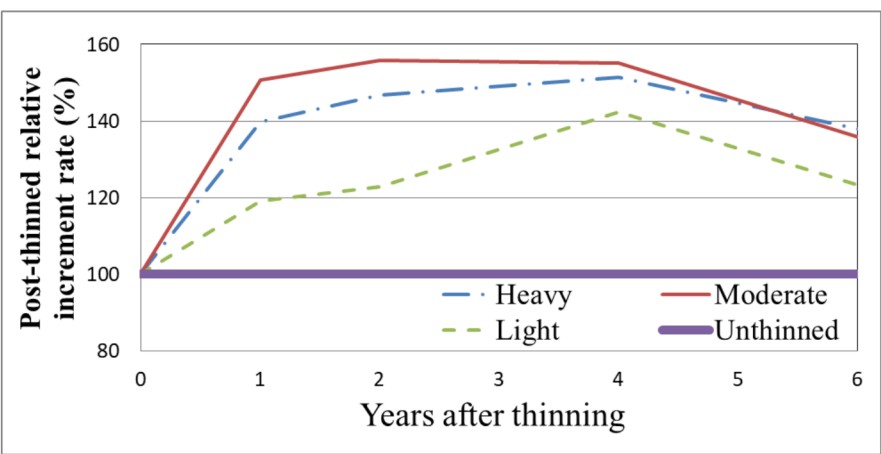

**Figure 5.** Percentage increases in the diameter at breast height (DBH) among different thinning treatments of Honduras mahogany in the Guanmiao Farm in Guanmiao, Taiwan after thinning relative to that of the un-thinned (relative increment of the un-thinned is 100%).

The results of the present study showed that under thinning of the lower layer, there were positive correlations between the DBH growth rate and the degree of thinning at one, two, four and six years after felling, consistent with the results of previous studies [8,15,34]. In addition, each treatment during the fourth year after felling resulted in an increase in DBH growth of 40% compared with that of the un-thinned, although this increase declined to ~20% by the sixth year after light thinning.

### 4. Conclusions and Suggestions

The present study examined the effect of thinning on the growth of 14-year-old Honduras mahogany trees in the Guanmiao area in Taiwan in which only one standard sample plot could be chosen for the basis of comparison of different thinning intensities, and the following conclusions were drawn:

(1) Thinning has a positive effect on promoting tree diameter growth. The growth rates of the retained small-diameter trees remain slow after felling, whereas medium-intensity thinning can increase the DBH of larger-diameter retained trees by a similar amount within a certain period.

(2) Increasing the thinning intensity can result in an increase in the average DBH growth rate six years after felling, whereas thinning of the lower layer can improve the diameter distribution. However, there is no significant change in the stand structure dynamics six years after felling.

(3) The periodic average growth rate of single tree DHB increased by up to 50% of the relative growth rate during the first year after harvesting and reached a relatively high rate during the second to fourth years after harvesting, following which it started to decline.

The results of lower layer thinning of mahogany plantations showed that all three thinning levels promoted DBH growth at four years after harvesting. In a forest production area, the implementation of intense thinning of the middle layer of poorly-formed trees can be implemented to obtain larger diameter trees with less effort. Alternatively, medium thinning can be used to maintain a higher stand size and limit the growth of branches.

A limitation of the present study is that observations were only conducted in the Guanmiao Farm in Guanmiao. Future studies should conduct observations in similar areas to increase the number of plot replicates. In addition, observations should be conducted in forest land near mountainous areas to determine whether there are significant differences in the time required to promote the growth of DBH under different thinning levels. The

results of the present study can act as a reference for the period of mahogany thinning in southern Taiwan.

**Author Contributions:** P.-H.P.: Conceptualization, Investigation, Writing—Original Draft. H.-L.P.: Conceptualization, Writing—Review & Editing. S.-L.T.: Data Curation, Formal analysis. C.-M.C.: Methodology, Validation. H.-L.C.: Data Curation, Formal analysis. Y.-X.Z.: Visualization, Data Curation. Y.-T.H.: Writing—Review & Editing. J.-C.C.: Project administration. All authors have read and agreed to the published version of the manuscript.

**Funding:** The research was supported by Council of Agriculture (102-13.1-FI-01 and 110 -11.1-FI-01).

**Institutional Review Board Statement:** Not applicable.

**Informed Consent Statement:** Not applicable.

**Data Availability Statement:** Data available on request from the authors.

**Acknowledgments:** Authors want to thanks to the Kagome Corporation, Taipei Forestry Specialist Association, Department of Forestry, Graduate Institute of Bioresources, National Pingtung University of Science and Technology, and the Taiwan Forestry Research Institute for help in field inventory and suggestions.

**Conflicts of Interest:** The authors reported no conflict of interest.

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
