# Peer review of "Effects of Thinning on the Growth and Relative Change in the Diameter of a Mahogany Plantation"

_forests, doi:10.3390/f13020213_

Round 1

Reviewer 1 Report

Dear Authors,

Please find my comments in the attached file

Regards

Author Response

Q1. Please clarify if replications were made or not

A1. It has been unified throughout the article, changing the words "control to un-thinned", "weak to light", and "strong to heavy".

Q2. Please check the instructions for authors how the descriptions of figures have to be formatted. If I am write size of text is 9 and instead of Fig. 2 Figure 2. in bold is written.

A2. The picture has been replaced. It has been unified throughout the article, the word "Fig. is changed to Figure", and the write size of text is 9.

Q3. Please check the instructions for authors how the descriptions of figures have to be formatted. If I am write size of text is 9 and instead of Fig. 2 Figure 2. in bold is written.

A3. It has been unified throughout the article, the word "Fig. is changed to Figure", and the write size of text is 9.

Q4. It would be nice to be constant in figure formatting. If previous figures are without yellow background, this ones has to be as well.

A4. The background of the picture has been changed to white, and the picture number has been corrected to "Figure 4".

Q5. Please fill these parts of the article.

A5. Supplementary descriptions have been modified in the new file.

Q6. Your reference formating is not in line with the Forests format.

A6. References in this archive are rearranged in the order they appear in the file.

Reviewer 2 Report

The article is improved in comparison to primary version, however I found one error to correct:

rows 127-128 (in previous version 99) - the sentence is still broken, with "and" at the end

Besides (general remark), style of citations should be changed, according to Instruction for Authors, i.e. with a use of numbers [1,2, etc.] not the names of cited Authors, because list of references is not in alphabetical order but in order as they appear in the text.

Author Response

Q1. rows 127-128 (in previous version 99) - the sentence is still broken, with "and" at the end

A1. The file has been modified as lines 188-189.。

Q2. Besides (general remark), style of citations should be changed, according to Instruction for Authors, i.e. with a use of numbers [1,2, etc.] not the names of cited Authors, because list of references is not in alphabetical order but in order as they appear in the text.

A2. References in this archive are rearranged in the order they appear in the file.

Reviewer 3 Report

The paper in its present form has some shortcomings and flaws that would have to be corrected before the paper could be consider for the acceptance. Detailed comments:

  • Pg.1, Title; Stand structure was not the object of the study. The title is necessary to take into account. Diameter structure was presented only. Please modify the title.
  • Pg.1., Abstract; Ln 31-33, the sentence is incomprehensible. What does in years 2-6 mean?
  • Pg.1, Ln 36; Key words; keywords do not repeat the title of the paper and/or it must be different from the article name.
  • Pg.2, Ln 66-67; Introduction – The authors in the bracket should be in ascending order of the years.
  • Pg.2, Ln 77; please correct …..16, 18, and 20, respectively.
  • Pg.2-4, Methods - I recommend to splitting-up as follows: study area, data collection, data analysis.
  • Pg.2, Ln 96; density was 1,718 trees ±242 ha-1?
  • Pg.3, Ln 104; ….thinning rate 9-30%.....out of which parameter? Specify it (number of trees, basal area or storage)?
  • Pg.3, Ln 105; …..on thinning of fallen trees and trees with broken or forked trunks…..? It´s a confusing. Fallen trees are dead ones – it´s not thinning. Thinnings are only applies to live trees. The same is valid for broken trunks. It´s desirable to divide it. Removed live trees = thinning. Fallen and broken trees = other decrease.
  • Pg.3, Ln 118-120; - it´s discussion not method – move it in this chapter.
  • Pg.3, Ln 122-126; …..it was assumed that trees with greater DBH…..what is the boundary (limiting) value? (12 cm, 15 cm or other)? Specify it.
  • Pg.4, Ln 127-132; Move it in the discussion chapter, it´s useless here.
  • Pg.4, Ln 147-149; Move it in the discussion chapter, it´s useless here. The authors in the bracket should be in ascending order of the years.
  • Pg.4, Result and Discussion; In some part is control, weak, moderate, strong thinning (treatment). In the Method (heavy, moderate, light, un-thinned) - It needs to be unified throughout the article.
  • Pg.11, Conclusion; Penultimate paragraph…….Last sentence. Alternatively, thinning can be used to maintain a higher stand size and limit diameter growth to improve wood quality. Wood quality was not evaluated in this study.
  • Pg.11 to 12; References - The authors mentioned in the text must be in accordance with the references.

Author Response

Q1. Pg.1, Title; Stand structure was not the object of the study. The title is necessary to take into account. Diameter structure was presented only. Please modify the title.

A1. Title revised to "Effects of thinning on the growth and relative change of in diameter of a mahogany plantation."

Q2. Pg.1., Abstract; Ln 31-33, the sentence is incomprehensible. What does in years 2-6 mean?

A2. 2-6 years represent the 2nd, 4th, and 6th years after thinning, which has been revised to "The mean periodic growth of single tree DBH after thinning was significantly different from that of the control group at tree ages of 16, 18, and 20 years” , as in line 31-33.

Q3. Pg.1, Ln 36; Key words; keywords do not repeat the title of the paper and/or it must be different from the article name.

A3. The keyword in the file has been modified to "Thinning intensities, DBH, Weibull distribution", as in line 36.

Q4. Pg.2, Ln 66-67; Introduction – The authors in the bracket should be in ascending order of the years.

A4. References in this archive are rearranged in the order they appear in the file.

Q5. Pg.2, Ln 77; please correct …..16, 18, and 20, respectively.

A5. The file has been modified as lines 74.

Q6. Pg.2-4, Methods - I recommend to splitting-up as follows: study area, data collection, data analysis.

A6. The file has been modified as lines 86, 95, and 121.

Q7. Pg.2, Ln 96; density was 1,718 trees ±242 ha-1?

A7. The file has been modified to read "the average number of mahogany plants", as in line 96-97.

Q8. Pg.3, Ln 104; ….thinning rate 9-30%.....out of which parameter? Specify it (number of trees, basal area or storage)?

A8. Here it means "number of trees", modified to "the planned thinning rate was 9–30% according to the number of trees for lower layer thinning.", as in lines 104-105.

Q9. Pg.3, Ln 105; …..on thinning of fallen trees and trees with broken or forked trunks…..? It´s a confusing. Fallen trees are dead ones – it´s not thinning. Thinnings are only applies to live trees. The same is valid for broken trunks. It´s desirable to divide it. Removed live trees = thinning. Fallen and broken trees = other decrease.

A9. Corrected as "oblique trees" in the file, as in line 106.

Q10. Pg.3, Ln 118-120; - it´s discussion not method – move it in this chapter.

A10. The file has been modified as in lines 188-193.

Q11. Pg.3, Ln 122-126; …..it was assumed that trees with greater DBH…..what is the boundary (limiting) value? (12 cm, 15 cm or other)? Specify it.

A11. In the file means "trees with DBH were relatively larger than other trees within the stand", modified as lines 123-125.

Q12. Pg.4, Ln 127-132; Move it in the discussion chapter, it´s useless here.

A12. The file has been modified as in lines 188-197.

Q13. Pg.4, Ln 147-149; Move it in the discussion chapter, it´s useless here. The authors in the bracket should be in ascending order of the years.

A13. The file has been modified as in lines 204-205. References in this archive are rearranged in the order they appear in the file.

Q14. Pg.4, Result and Discussion; In some part is control, weak, moderate, strong thinning (treatment). In the Method (heavy, moderate, light, un-thinned) - It needs to be unified throughout the article.

A14. It has been unified throughout the article, changing the words "control to un-thinned", "weak to light", and "strong to heavy".

Q15. Pg.11, Conclusion; Penultimate paragraph…….Last sentence. Alternatively, thinning can be used to maintain a higher stand size and limit diameter growth to improve wood quality. Wood quality was not evaluated in this study.

A15. The file has been modified to "limit the growth of branches" as in line 312-313.

Q16. Pg.11 to 12; References - The authors mentioned in the text must be in accordance with the references.

A16. References in this archive are rearranged in the order they appear in the file.

This manuscript is a resubmission of an earlier submission. The following is a list of the peer review reports and author responses from that submission.

Round 1

Reviewer 1 Report

The paper includes an interesting results of thinning experiment performed in 14-years-old stands of Honduras mahogany introduced in Taiwan. Introduction is sufficient but in the chapter of study area, materials and methods some parts are not satisfactory (see the file with detailed remarks). The weakness of the experiment is lack of replications as well as relatively small research units (0,05 ha each) but it is improved as much as possible with a rich set of statistical tools. Results are well described but the discussion is relatively poor, maybe because of small number of related papers. The "mixture" of results and discussion in one chapter is acceptable but it makes the paper more difficult in perception. Final conclusions should be well explained corresponding to related parts of chapter 2 (see the file). The work can be published after minor revision, especially chapters 2 and 4.

Reviewer 2 Report

General comments

The work being assessed has many serious drawbacks and should not, in my opinion, be published in the Forests journal. One of the drawbacks of the work is the lack of a clearly defined and properly argued scientific problem. The introduction was limited only to the description of the ecological and economic (social) significance of the analyzed tree species. There is no broader reference to the concepts defined in the title of the work, i.e. tree thinning, structure and growth. The introduction does not contain a description of the current knowledge on these issues, nor does it indicate knowledge gaps that should be filled. The defined goal of the work is of a technical rather than scientific nature.

The second, much more serious drawback, which in my opinion disqualifies this work, is the incorrect design of the experiment used. The main task of the arrangement of the experiment is to eliminate or limit the influence of factors other than the analyzed one, it is especially important in field experiments, in which the researcher has no possibility to directly control the conditions of the experiment (spatial differentiation of the environment conditions, as well as heterogeneity of the tested plant communities, e.g. genetic, resulting from various treatments performed previously - planting spacing, cleaning cuts, etc.). The use of one and, in addition, extremely small (0.05 ha) sample plot representing each treatment, seems to be an insufficient experiment plan authorizing to conclude about the impact of the studied treatment. Moreover, the ambiguous description of the sample plots used, shows that at the time of the treatment they differed in the structure and density of trees. In my opinion, the applied experiment plan (one small plot with one treatment) does not allow for more general conclusions, at most the observations made can be treated as preliminary results justifying further, much more correct planning of the experiment. In the case of modeling the dbh structure of the stand using the Weibull distribution, the authors should use proven methods of model construction based on prediction or parameter recovery.